# On Dynamical Measures of Quantum Information

**DOI:** 10.3390/e27040331

**Published:** 2025-03-21

**Authors:** James Fullwood, Arthur J. Parzygnat

**Affiliations:** 1School of Mathematics Statistics, Hainan University, 58 Renmin Avenue, Haikou 570228, China; 2Experimental Study Group, Massachusetts Institute of Technology, Cambridge, MA 02139, USA; arthurjp@mit.edu

**Keywords:** quantum information, entropy, mutual information

## Abstract

In this work, we use the theory of quantum states over time to define joint entropy for timelike-separated quantum systems. For timelike-separated systems that admit a dual description as being spacelike-separated, our notion of entropy recovers the usual von Neumann entropy for bipartite quantum states and thus may be viewed as a spacetime generalization of von Neumann entropy. Such an entropy is then used to define dynamical extensions of quantum joint entropy, quantum conditional entropy, and quantum mutual information for systems separated by the action of a quantum channel. We provide an in-depth mathematical analysis of such information measures and the properties they satisfy. We also use such a dynamical formulation of entropy to quantify the information loss/gain associated with the dynamical evolution of quantum systems, which enables us to formulate a precise notion of information conservation for quantum processes. Finally, we show how our dynamical entropy admits an operational interpretation in terms of quantifying the amount of state disturbance associated with a positive operator- valued measurement.

## 1. Introduction

The lack of a joint quantum state associated with timelike-separated quantum systems yields an obstacle to generalizing various aspects of classical probability to the quantum domain. In particular, while classical measures of information such as joint entropy, conditional entropy, and mutual information associated with a pair of random variables (X,Y) are defined irrespective of the spacetime separation of *X* and *Y*, their quantum counterparts are only defined for spacelike-separated systems. This is due to the fact that, unlike timelike-separated systems, the information contained in spacelike-separated systems *A* and *B* is encoded by a density operator $\rho_{AB}$, which one may utilize to define fundamental measures of quantum information. For example, the quantum joint entropy, quantum conditional entropy, and quantum mutual information associated with spacelike-separated systems *A* and *B* are given by(1)$$\begin{array}{ccc} \underline{\mathrm{Quantum}\;\mathrm{Joint}\;\mathrm{Entropy}}:\;\;S(A\;,B) & = & S(\rho_{AB})\;; \end{array}$$(2)$$\begin{array}{ccc} \underline{\mathrm{Quantum}\;\mathrm{Conditional}\;\mathrm{Entropy}}:\;\;H(B|A) & = & S\left(\rho_{AB}\right)-S\left(\rho_{A}\right)\;; \end{array}$$(3)$$\begin{array}{ccc} \underline{\mathrm{Quantum}\;\mathrm{Mutual}\;\mathrm{Information}}:\;I(A:B) & = & S\left(\rho_{A}\right)+S\left(\rho_{B}\right)-S\left(\rho_{AB}\right)\;, \end{array}$$
where S(·) denotes the von Neumann entropy and $\rho_{A}$ and $\rho_{B}$ are the reduced density operators of $\rho_{AB}$. But what if the systems *A* and *B* are instead timelike-separated? There should certainly exist an analog of such information measures in a scenario where $\rho_{A}$ dynamically evolves into $\rho_{B}$.

While various proposals for dynamical extensions of quantum information measures exist [1,2,3,4,5,6,7,8,9,10], in this work, we employ the quantum state over time formalism as first introduced in Ref. [11] to define dynamical extensions of the information measures given by Equations (1)–(3). In doing so, we show that one may extend the above measures of quantum information to the dynamical setting in such a way that recovers the associated classical information measures for states that undergo decoherence with respect to an orthonormal basis corresponding to the eigenstates of some observable, and for which the associated dynamics are dephasing with respect to such a basis. For timelike-separated systems that admit a dual description as being spacelike-separated, our information measures coincide with the information measures given by Equations (1)–(3) and hence resemble the principle of general covariance in relativity. We also show how our approach naturally yields a quantum generalization of the notion of information loss associated with stochastic processes as defined in Refs. [12,13], which naturally lends itself to a precise quantitative notion of conservation of quantum information for open quantum systems (cf. Remark 5).

A quantum state over time is a bipartite Hermitian operator $\varrho_{AB}$ of unit trace associated with a quantum state $\rho_{A}$ that dynamically evolves under the action of a quantum channel E:S(A)→S(B), where S(·) denotes the space of states of a system. Although the marginals of such a quantum state over time $\varrho_{AB}$ are the density operators $\rho_{A}$ and $\rho_{B}=E\left(\rho_{A}\right)$, in contrast to bipartite density operators, a quantum state over time is not positive in general and thus is not a quantum state in the traditional sense (hence the notation $\varrho_{AB}$ as opposed to $\rho_{AB}$). As such, extending the density operator formalism of quantum theory to include non-positive quantum states over time is akin to the extension of the geometry of space to the geometry of spacetime, where the spacetime metric exhibits a Lorentzian (as opposed to Euclidean) signature as it extends over time.

Quantum states over time have been derived from simple physical assumptions in Refs. [14,15,16] and have been shown to encompass various notions of a spatiotemporal state associated with timelike-separated quantum systems [17], such as the transition matrices of Nakata et al. [18]; the causal states of Leifer and Spekkens [19,20,21]; the pseudo-density operators of Fitzsimons, Jones, and Vedral [14,22]; the 2-states of Watanabe, Aharanov, and Reznik [23,24,25]; the Wigner-function approach of Wootters [11,26]; and the compound states of Ohya [27]. Although there exist alternative approaches to spatiotemporal quantum states, such as the superdensity operators of Cotler, Jian, Qi, and Wilczek [28] and the process matrices of Oreshkov, Costa, and Brukner [29], such approaches require algebras that are twice the dimension of the algebras required for the construction of quantum states over time. Moreover, a dynamical quantum Bayes’ rule formulated in terms of quantum states over time has led to the discovery of time-symmetric correlations for open quantum systems that lose information to their environment, revealing new insights into the nature of reversibility for interacting systems [30].

As quantum states over time are not positive in general, one must extend the notion of von Neumann entropy to unit trace Hermitian operators in order to define a notion of entropy associated with quantum states over time. While there has been recent work making use of various extensions of von Neumann entropy to non-positive operators—such as an approach using an analytic continuation of the logarithm [18,31] and an approach using the singular values of a Hermitian operator [32]—we find that such approaches do not yield results one would expect from a quantum generalization of the classical theory of dynamical information, such as the vanishing of conditional entropy under deterministic evolution [33]. The aforementioned approaches introduced such generalizations with specific applications in mind, as opposed to systematically constructing one based on physical and mathematical principles. In the approach taken here, however, we are guided by the assumption that if a state undergoes trivial dynamics corresponding to the identity channel, then the entropy of the associated quantum state over time should coincide with the entropy of the initial state, as no information is gained or lost in such a process. For those familiar with the language of categories [34,35], our approach follows the mathematical philosophy that invariants of objects in a category should be viewed as more general invariants of the morphisms specialized to identity maps.

In terms of the functional calculus, we then employ the extension of von Neumann entropy to non-positive (normal) operators given by(4)$$S\left(X\right)=-\mathrm{tr}\left(X\log\sqrt{X^{†}X}\right)\;.$$While the operational meaning of the entropy functional (Equation 4) is not yet well understood, it has been utilized in the context of non-Hermitian physics associated with quantum many-body systems, non-unitary conformal field theories, and processes involving post-selection [8,36,37]. Moreover, the Page curve [38,39,40] for non-Hermitian systems was recently computed in Ref. [37] using precisely the entropy functional (Equation 4). However, it remained an open question in all of these works to analyze the properties of (Equation 4) in greater detail. This is what our present paper achieves. Unlike extensions of von Neumann entropy based on the analytic continuation or singular values, we show that such a notion of entropy retains more properties of von Neumann entropy than its counterparts, such as strong convex linearity (cf. item iv. of Proposition 1). Interestingly, while the entropy functional (Equation 4) does not in general satisfy subadditivity, it does seem to satisfy subadditivity when restricted to quantum states over time. In particular, in all known examples we find(5)$$S\left(\varrho_{AB}\right)\leq S\left(\rho_{A}\right)+S\left(\rho_{B}\right)\;,$$
where $\varrho_{AB}$ is the quantum state over time associated with a dynamical evolution of $\rho_{A}$ into $\rho_{B}=E\left(\rho_{A}\right)$ for some quantum channel E:S(A)→S(B). As such, our calculations suggest a numerical test for the temporal compatibility of non-positive operators [41,42], namely, that if a bipartite Hermitian operator of unit trace does not in fact satisfy subadditivity with respect to the entropy functional (Equation 4), then it may not be realized as a quantum state over time.

The notion of a quantum state over time together with the extension of von Neumann entropy given by (Equation 4) then enables one to define dynamical extensions of the quantum information measures given by (Equation 1) in such a way that not only retains many of the fundamental properties of such information measures in the spatial case, but also in a way that recovers the classical information measures for states that have undergone decoherence. In what follows, we investigate properties of such dynamical measures of quantum information from both a mathematical and physical perspective, establishing various results along the way.

## 2. The Entropy of a Quantum Process

Let *A* and *B* denote quantum systems that correspond to the input and output of a quantum channel E:A→B, which is defined to be a completely positive trace-preserving (CPTP) map between the algebras A and B of linear operators on the associated Hilbert spaces $H_{A}$ and $H_{B}$, respectively. We assume that the systems *A* and *B* are finite-dimensional. More precisely, we assume that A and B are isomorphic to full matrix algebras. The set of states on *A* and *B* will be denoted by S(A) and S(B), which consist of all density operators on $H_{A}$ and $H_{B}$, respectively.

Given an initial state ρ∈S(A) for the channel E:A→B, we refer to the pair (E,ρ) as a ***process***, as it encapsulates the dynamical evolution of the state ρ into the state E(ρ) via the channel E. The ***quantum state over time*** associated with the process (E,ρ) is the bipartite operator E⋆ρ∈A⊗B given by(6)$$E⋆\rho =\frac{1}{2}\left\{\rho ⊗1\;,J[E]\right\}\;,$$
where {·,·} denotes the anti-commutator and J[E] is the ***Jamiołkowski*** matrix of the channel E [43], which is given by$$J\left[E\right]=\sum_{i,j}\left|i\rangle \langle j\right|⊗E(|j\rangle \langle i|)\;.$$Although the quantum state over time E⋆ρ is Hermitian and of unit trace, it is not positive in general. However, as ${\mathrm{tr}}_{B}\left(E⋆\rho \right)=\rho$ and ${\mathrm{tr}}_{A}\left(E⋆\rho \right)=E\left(\rho \right)$, the quantum state over time E⋆ρ preserves the marginal states corresponding to the input and output of the channel E. The negative eigenvalues of a non-positive state over time E⋆ρ then serve as a witness to temporal correlations arising as a result of the dynamical process (E,ρ), which have no spatial analog. In Refs. [14,15,16], the quantum state over time E⋆ρ was derived under physically motivated hypotheses such as hermiticity, preservation of the marginal states, and linearity with respect to the initial state ρ. While there are other possible constructions of quantum states over time that may be arrived at by relaxing some of these hypotheses, it follows from a no-go theorem proved in Ref. [44] that the hypotheses of preservation of marginal states and linearity with respect to ρ forces quantum states over time to be non-positive in general (a similar no-go theorem was obtained in Ref. [14]). Thus, the non-positivity of quantum states over time is inevitable for general processes (E,ρ).

Equipped with quantum states over time, we define the ***entropy*** of a process (E,ρ) to be the real number S(E,ρ) given by(7)$$S\left(E,\rho \right)=-\mathrm{tr}\left((E⋆\rho )\log|E⋆\rho |\right)\;,$$
where $\left|E⋆\rho \right|=\sqrt{{\left(E⋆\rho \right)}^{†}\left(E⋆\rho \right)}$. For processes (E,ρ) such that E⋆ρ is positive, the entropy S(E,ρ) is simply the von Neumann entropy of the quantum state over time E⋆ρ. Thus, S(E,ρ) for general processes may be viewed as a direct generalization of von Neumann entropy to non-positive quantum states over time. We note that the entropy S(E,ρ) is not to be confused with the entropy *exchange* associated with the process (E,ρ), which is often denoted by S(ρ,E) [45].

A primary justification for our definition of entropy for quantum processes is provided by the following:

**Theorem** **1.**
*Given a quantum system A with associated algebra A, let I:A→A denote the identity channel on A. Then, for every state ρ∈S(A),*

(8)
S(I,ρ)=S(ρ),

*where S(ρ) is the von Neumann entropy of ρ.*


Before proving Theorem 1, we introduce some notation. Given an operator *X* on a finite-dimensional Hilbert space H, the ***multi-spectrum*** of *X* is the multi-set mspec(X) associated with the eigenvalues of *X* and their multiplicities. Thus, mspec(X) consists of $\mathrm{tr}(1_{H})$ elements, with repetitions corresponding to the multiplicities of the eigenvalues of *X*. For example, $\mathrm{mspec}\left(1_{H}\right)=\left\{1,\ldots ,1\right\}$.

**Proof** **of** **Theorem** **1.**Let ρ∈S(A), let I:A→A denote the identity channel, and suppose $\mathrm{mspec}\left(\rho \right)=\left\{\lambda_{1},\ldots ,\lambda_{n}\right\}$, so that $n=\mathrm{tr}(1_{H_{A}})$. Since I⋆ρ is Hermitian, it is diagonalizable. Hence,(9)$$S\left(I,\rho \right)=-\mathrm{tr}\left(I⋆\rho \log|I⋆\rho |\right)=-\sum_{\lambda \in \mathrm{mspec}(I⋆\rho )}\lambda \log\left|\lambda \right|\;.$$By Lemma A5 in Appendix A, it follows that(10)$$\mathrm{mspec}\left(I⋆\rho \right)=\left\{\lambda_{1},\ldots ,\lambda_{n}\right\}\cup \left.\left\{\pm \frac{\lambda_{i}+\lambda_{j}}{2}\;\;\right|\;\;i,j\in \left\{1,\ldots ,n\right\}\;\;\mathrm{with}\;i\neq j\right\}\;,$$
and since the function f(x)=xlog|x| is an odd function, it follows from (Equation 9) that the multi-set$$\left.\left\{\pm \frac{\lambda_{i}+\lambda_{j}}{2}\;\;\right|\;\;i,j\in \left\{1,\ldots ,n\right\}\;\;\mathrm{with}\;i\neq j\right\}$$
has 0 net contribution to S(I,ρ). It then follows that$$S\left(I,\rho \right)=-\sum_{i=1}^{n}\lambda_{i}\log\left|\lambda_{i}\right|=S\left(\rho \right)\;,$$
as desired. □

In light of Theorem 1, one may view the entropy S(E,ρ) as a generalization of von Neumann entropy from states to processes that recovers the von Neumann entropy of states via processes modeled by the identity channel. Certainly, such a property is natural as the identity channel does not physically alter the state ρ in any way. Moreover, such a property also holds at the classical level for Shannon entropy, as the joint entropy associated with the pair (X,X), where *X* is a classical random variable, is simply the Shannon entropy H(X).

To conclude this section, we recall three alternative approaches to extending von Neumann entropy to non-positive operators and show that in each case, an associated analog of Theorem 1 does *not* hold. This provides further justification for our use of the entropy functional S(X)=−tr(Xlog|X|) over other alternative extensions of von Neumann entropy.

**Pseudo-Entropy.** In Ref. [31], von Neumann entropy is extended to non-positive operators by using the analytic continuation of the logarithm, which is referred to as *pseudo-entropy*. If we wish to apply such an approach to quantum states over time, we must employ an analytic continuation of the logarithm on a domain that contains the negative real axis, as quantum states over time admit negative real eigenvalues in general. So now, let log(z) be an analytic continuation of the logarithm for *z* on a domain that contains the negative real axis, and define an associated entropy for processes (E,ρ) via the formula$$S_{\mathrm{PE}}\left(E,\rho \right)=-\mathrm{tr}\left(\left(E⋆\rho \right)\log\left(E⋆\rho \right)\right)\;,$$
where the subscript PE stands for pseudo-entropy. If ρ∈S(A) is a density operator representing a qubit in a pure state and I:A→A denotes the identity channel, then it follows from (Equation 10) that the state over time I⋆ρ has eigenvalues 0, 1, and ±1/2. We then have $$\begin{array}{cc} S_{\mathrm{PE}}\left(I,\rho \right) & =-\mathrm{tr}\left(\left(I⋆\rho \right)\log\left(I⋆\rho \right)\right)\\ \; & =-1\log\left(1\right)-0\log\left(0\right)-\frac{1}{2}\log\left(\frac{1}{2}\right)-\left(-\frac{1}{2}\right)\log\left(-\frac{1}{2}\right)\\ \; & =-\frac{1}{2}\left(\log\left|\frac{1}{2}\right|+i\mathrm{Arg}\left(\frac{1}{2}\right)\right)+\frac{1}{2}\left(\log\left|-\frac{1}{2}\right|+i\mathrm{Arg}\left(-\frac{1}{2}\right)\right)\\ \; & =\frac{i}{2}\left(\mathrm{Arg}\left(-\frac{1}{2}\right)-\mathrm{Arg}\left(\frac{1}{2}\right)\right)\\ \; & =\frac{i\pi }{2}\\ \; & \neq S(\rho )\;, \end{array}$$
where the final inequality follows from the fact that S(ρ)=0 since ρ is a pure state. Therefore, an analog of Theorem 1 does not hold for pseudo-entropy.

**SVD entropy.** In Ref. [32], an extension of von Neumann entropy was introduced, which was referred to as *SVD entropy* (where SVD is an acronym for singular value decomposition). Given a square matrix *X*, its SVD entropy is the real number $S_{\mathrm{SVD}}\left(X\right)$ given by$$S_{\mathrm{SVD}}\left(X\right)=S\left(\frac{\sqrt{X^{†}X}}{\mathrm{tr}(\sqrt{X^{†}X})}\right)\;,$$
where S(·) denotes von Neumann entropy. Using the SVD entropy, we can then define an entropy for processes by the formula$$S_{\mathrm{SVD}}\left(E,\rho \right)=S_{\mathrm{SVD}}\left(E⋆\rho \right)\;.$$If ρ∈S(A) is a pure state of a single qubit and I:A→A is the identity channel, then the eigenvalues of the operator$$\frac{1}{\mathrm{tr}(I⋆\rho )}\sqrt{{\left(I⋆\rho \right)}^{†}\left(I⋆\rho \right)}$$
are 0, 1/2, 1/4, and 1/4. Thus,$$S_{\mathrm{SVD}}\left(I,\rho \right)=\log\left(2\sqrt{2}\right)\neq 0=S\left(\rho \right)\;.$$Therefore, an analog of Theorem 1 does not hold for SVD entropy.

**Extending von Neumann entropy via** −|x|log|x|. In Ref. [3], the von Neumann entropy is extended to self-adjoint matrices using the formula(11)$$\tilde{S}\left(X\right)=-\mathrm{tr}\left(|X|\log|X|\right)\;,$$
where $\left|X\right|=\sqrt{X^{†}X}$. We can then use $\tilde{S}$ to define the entropy of processes by the formula$$\tilde{S}\left(E,\rho \right)=\tilde{S}\left(E⋆\rho \right)\;.$$If ρ∈S(A) represents a qubit in a pure state and I:A→A is the identity channel, then the operator |I⋆ρ| has eigenvalues 0, 1, 1/2, and 1/2. Hence,$$\tilde{S}\left(I,\rho \right)=\log\left(2\right)\neq S\left(\rho \right)\;.$$Therefore, an analog of Theorem 1 does not hold for the entropy functional from (Equation 11).

## 3. Mathematical Properties of the Entropy Functional

In this section, we investigate mathematical properties of the entropy functional on Hermitian matrices given by S(X)=−tr(Xlog|X|). As for notation, we let $H_{n}$ denote the set of all n×n Hermitian matrices, and we let $H={⋃}_{n=1}^{\infty }H_{n}$. The set consisting of unit-trace elements in $H_{n}$ will be denoted by $Q_{n}$, and elements of $Q_{n}$ will be referred to as ***quasi-states***. We first define the entropy functional on arbitrary hermitian matrices, which was employed earlier in our definition of entropy for quantum processes. Figure 1 shows the graph of this entropy function for diagonal quasi-states in dimensions 2 and 3.

**Definition** **1.***The* **entropy** *of a self-adjoint matrix $X\in H_{n}$ is the real number S(X) given by*(12)$$S\left(X\right)=-\mathrm{tr}\left(X\log\left|X\right|\right)\equiv -\sum_{\lambda \in \mathrm{mspec}(X)}\lambda \log\left|\lambda \right|.$$*If X is a density matrix, then S(X) will be referred to as the ***von Neumann entropy** 
*of X.*

In the following proposition, we state and prove various mathematical properties of the entropy functional given by (Equation 12), all of which may be viewed as direct generalizations of properties satisfied by von Neumann entropy. We note that the final property listed is a generalization of the Fannes–Audenaert inequality [46] to the entropy of Hermitian matrices as given by (Equation 12).

**Proposition** **1.***The entropy functional S:H→R given by *(Equation 12)* satisfies the following properties.*
*i.* Extension: *S(X) is the von Neumann entropy of X for every density matrix X.**ii.* Unitary Invariance: *$S\left(X\right)=S\left(UXU^{†}\right)$ for every unitary U and for all X∈H.**iv.* Additivity: *S(X⊗Y)=S(X)+S(Y) for all quasi-states $X\in Q_{n}$ and $Y\in Q_{m}$.**ii.* Strong Convex Linearity: *If p:{1,…,k}→R is a quasi-probability distribution (so that $\sum_{i}p_{i}=1$) and ${\left\{X_{i}\right\}}_{i=1}^{k}\subset Q_{n}$ is a collection of mutually orthogonal quasi-states, then*(13)$$S\left(\sum_{i=1}^{k}p_{i}X_{i}\right)=H\left(p\right)+\sum_{i=1}^{k}p_{i}S\left(X_{i}\right)\;.$$*v.* Continuity: *The entropy function S is continuous. Moreover, if $\rho ,\xi \in Q_{n}$ have multispectrums $\left\{\lambda_{i}\right\}$ and $\left\{\mu_{i}\right\}$ such that $\lambda_{i}\mu_{i}\geq 0$ for all i∈{1,⋯,n}, and if ${∥\rho -\xi ∥}_{1}\leq 1/e$, then*(14)$$|S\left(\rho \right)-S\left(\xi \right)|\leq {∥\rho -\xi ∥}_{1}\log\left(n\right)+\eta \left({∥\rho -\xi ∥}_{1}\right),$$*where η(x)=−xlog(x).*


**Proof.** We prove each of the properties one at a time.
The statement follows from the fact that a density matrix *X* is positive and therefore satisfies X=|X|.The statement follows from the cyclicity of the trace and the functional calculus for matrices, or equivalently, from the fact that $UXU^{†}$ has the same eigenvalues as *X*.Suppose $\mathrm{mspec}\left(X\right)=\left\{\lambda_{i}\right\}$ and $\mathrm{mspec}\left(Y\right)=\left\{\mu_{j}\right\}$, so that $\mathrm{mspec}\left(X⊗Y\right)=\left\{\lambda_{i}\mu_{j}\right\}$. Then,$$\begin{array}{ccc} S(X⊗Y) & = & -\sum_{i,j}\lambda_{i}\mu_{j}\log\left|\lambda_{i}\mu_{j}\right|=-\sum_{i,j}\lambda_{i}\mu_{j}\left(\log\left|\lambda_{i}\right|+\log\left|\mu_{j}\right|\right)\\ & = & -\sum_{i,j}\lambda_{i}\mu_{j}\log\left|\lambda_{i}\right|-\sum_{i,j}\lambda_{i}\mu_{j}\log\left|\mu_{j}\right|=-\sum_{i}\lambda_{i}\log\left|\lambda_{i}\right|-\sum_{j}\mu_{j}\log\left|\mu_{j}\right|\\ & = & S(X)+S(Y), \end{array}$$
as desired.Let $\left\{\lambda_{j}^{i}\right\}$ denote the multispectrum of $X_{i}$ for all i∈{1,…,k}, so that the multispectrum of $\sum_{i=1}^{k}p_{i}X_{i}$ is $\left\{p_{i}\lambda_{j}^{i}\right\}$, where j∈{1,…,n}. Then,$$\begin{array}{ccc} S\left(\sum_{i=1}^{k}p_{i}X_{i}\right) & = & -\sum_{i=1}^{k}\sum_{j=1}^{n}p_{i}\lambda_{j}^{i}\log\left|p_{i}\lambda_{j}^{i}\right|=-\sum_{i=1}^{k}\sum_{j=1}^{n}p_{i}\lambda_{j}^{i}\log\left|p_{i}\right|-\sum_{i=1}^{k}\sum_{j=1}^{n}p_{i}\lambda_{j}^{i}\log\left|\lambda_{j}^{i}\right|\\ & = & -\sum_{i=1}^{k}p_{i}\log\left|p_{i}\right|\sum_{j=1}^{n}\lambda_{j}^{i}-\sum_{i=1}^{k}p_{i}\sum_{j=1}^{n}\lambda_{j}^{i}\log\left|\lambda_{j}^{i}\right|=H\left(p\right)+\sum_{i=1}^{k}p_{i}S\left(X_{i}\right), \end{array}$$
as desired.Let ${\tilde{\eta }}_{n}:R^{n}\to R$ be the function given by$${\tilde{\eta }}_{n}\left(x_{1},\cdots ,x_{n}\right)=-\sum_{i=1}^{n}x_{i}\log\left|x_{i}\right|$$(where we set 0log0=0), and let $E_{n}:Q_{n}\to R^{n}$ be the function given by$$E_{n}\left(\rho \right)=\left(\lambda_{1},\cdots ,\lambda_{n}\right),$$
where $\mathrm{mspec}\left(\rho \right)=\left\{\lambda_{1},\cdots ,\lambda_{n}\right\}$. The identity $S\left(\rho \right)=\left({\tilde{\eta }}_{n}\circ E_{n}\right)\left(\rho \right)$ for all $\rho \in Q_{n}$ shows that *S* is the composite of two continuous functions, from which it follows that *S* is continuous as well (cf. Chapters 1 and 5 in Ref. [47]).To prove the Fannes-type inequality (Equation 14), we adapt the standard proof for density matrices to the case at hand (cf. Theorem 11.6 of Ref. [45]). Let η:[0,∞)→R be the function given by η(x)=−xlog(x), and let $\tilde{\eta }:R\to R$ be the odd completion of η, so that $\tilde{\eta }\left(x\right)=-x\log\left|x\right|$. Suppose now that $\rho ,\xi \in Q_{n}$ are unit trace elements with multispectrums $\left\{\lambda_{i}\right\}$ and $\left\{\mu_{i}\right\}$, suppose $\lambda_{i}\mu_{i}\geq 0$ for all i∈{1,⋯,n}, and also suppose ${∥\rho -\xi ∥}_{1}\leq 1/e$, so that $|\lambda_{i}-\mu_{i}|<\frac{1}{2}$. Then,$$\begin{array}{ccc} \left|S(\rho )-S(\xi )\right| & = & \left|\sum_{i=1}^{n}\tilde{\eta }\left(\lambda_{i}\right)-\tilde{\eta }\left(\mu_{i}\right)\right|\leq \sum_{i=1}^{n}|\tilde{\eta }\left(\lambda_{i}\right)-\tilde{\eta }\left(\mu_{i}\right)|\\ & = & \sum_{i=1}^{n}|\eta \left(|\lambda_{i}|\right)-\eta \left(|\mu_{i}|\right)|\leq \sum_{i=1}^{n}\eta \left(\left|\;\right|\lambda_{i}\left|-\right|\mu_{i}\left|\;\right|\right)\\ & = & \sum_{i=1}^{n}\eta \left(|\lambda_{i}-\mu_{i}|\right), \end{array}$$
where the second equality follows from the fact that rs≥0 implies$$|\tilde{\eta }\left(r\right)-\tilde{\eta }\left(s\right)|=|\eta \left(|r|\right)-\eta \left(|s|\right)|,$$
and the second inequality follows from the fact that $|r-s|<\frac{1}{2}$ with *r* and *s* non- negative implies$$|\eta \left(r\right)-\eta \left(s\right)|\leq \eta \left(|r-s|\right).$$Now, set $\epsilon_{i}=\left|\lambda_{i}-\mu_{i}\right|$ for all i∈{1,⋯,n}, and set $\epsilon =\epsilon_{1}+\cdots +\epsilon_{n}$, so that $\epsilon \leq {∥\rho -\xi ∥}_{1}$. We then have$$\begin{array}{ccc} \left|S(\rho )-S(\xi )\right| & \leq & \sum_{i=1}^{n}\eta \left(\left|\lambda_{i}-\mu_{i}\right|\right)=\sum_{i=1}^{n}\eta \left(\epsilon_{i}\right)=\sum_{i=1}^{n}\left(\epsilon \eta \left(\epsilon_{i}/\epsilon \right)-\epsilon_{i}\log\left(\epsilon \right)\right)\\ & = & \epsilon \left(\sum_{i=1}^{n}\eta \left(\epsilon_{i}/\epsilon \right)\right)+\eta \left(\epsilon \right)\leq {∥\rho -\xi ∥}_{1}\log\left(n\right)+\eta \left({∥\rho -\xi ∥}_{1}\right)\;, \end{array}$$
where the final inequality follows from the fact that $\epsilon \leq {∥\rho -\xi ∥}_{1}$ and η is monotone-increasing on [0,1/e], thus concluding the proof. □


**Remark** **1** (Failure of subadditivity for quasi-states).
*If $\rho_{AB}\in A⊗B$ is a density matrix with marginals $\rho_{A}={\mathrm{tr}}_{B}\left(\rho_{AB}\right)$ and $\rho_{B}={\mathrm{tr}}_{A}\left(\rho_{AB}\right)$, then*

$$S\left(\rho_{AB}\right)\leq S\left(\rho_{A}\right)+S\left(\rho_{B}\right)\;,$$

*which is a property of von Neumann entropy commonly referred to as subadditivity. However, this property does not hold in general for the extended entropy function evaluated on arbitrary quasi-states whose marginals are states. For a counter-example, consider the family of non-positive Hermitian matrices*

$$\varrho_{AB}=\frac{1-\omega }{12}\left(\begin{array}{cccc} -6 & \sqrt{5} & \sqrt{5} & 0\\ \sqrt{5} & 8 & 0 & \sqrt{5}\\ \sqrt{5} & 0 & 8 & \sqrt{5}\\ 0 & \sqrt{5} & \sqrt{5} & 2 \end{array}\right)+\frac{\omega }{4}1_{4}$$

*parameterized by ω∈[0,1]. The marginals of $\varrho_{AB}$ are given by the same reduced density matrix*

$$\rho_{A}=\frac{1-\omega }{6}\left(\;\begin{array}{cc} 1 & \sqrt{5}\\ \sqrt{5} & 5 \end{array}\right)+\frac{\omega }{2}1_{2}=\rho_{B}.$$

*In such a case, $\rho_{A}$ and $\rho_{B}$ are pure states when ω=0 and maximally mixed states when ω=1. The entropies of $\rho_{A}$ and $\rho_{B}$ then start at zero for ω=0 and are monotonically increasing as ω goes from 0 to 1. Meanwhile, $S(\varrho_{AB})\approx 0.29$ when ω=0, illustrating that $S\left(\varrho_{AB}\right)>S\left(\rho_{A}\right)+S\left(\rho_{B}\right)$ for ω=0, thus violating subadditivity. Graphs of $S(\varrho_{AB})$ and $S\left(\rho_{A}\right)+S\left(\rho_{B}\right)$ as functions of ω are shown in Figure 2.*

*However, $\varrho_{AB}$ is not a quantum state over time for all values of ω∈[0,1]. To see this, note that $\varrho_{AB}$ is a quantum state over time if and only if there exists a completely positive trace-preserving map E:A→B such that*

(15)
$$\varrho_{AB}=\frac{1}{2}\left\{\rho_{A}⊗1\;,J\left[E\right]\right\}\;.$$

*Taking the partial transpose of both sides of (Equation 15), we arrive at the equation*

$$\varrho_{AB}=\frac{1}{2}\left\{\rho_{A}⊗1\;,C\left[E\right]\right\}\;,$$

*where C[E] is the Choi matrix of E [48]. This equation is an example of a Sylvester–Lyapunov equation (cf. Refs. [49,50,51,52]), which admits a unique solution for the matrix C[E] for all values of ω∈(0,1]. Now, since E is completely positive if and only if C[E] is positive, we then plot the eigenvalues of C[E] as functions of ω in Figure 2, in which case we find that C[E] has negative eigenvalues for all ω∈[0,1]. As such, $\varrho_{AB}$ is not a quantum state over time. At present, we do not know of any examples of quantum states over time that violate subadditivity, which leads us to conjecture that the entropy functional S restricted to quantum states over time does in fact satisfy subadditivity. We will provide further support for this conjecture in the context of dynamical mutual information in Section 5.*


**Remark** **2.***Refs. [8,36,37] make use of the entropy functional *(Equation 12)* for different purposes, which we briefly discuss. In Ref. [8], the entropy functional is used as an extension of von Neumann entropy to 2-states [23,24,25], which are also referred to as transition matrices in Ref. [18]. As shown in Ref. [17], 2-states/transition matrices may be viewed as non-Hermitian quantum states over time whose real part is the Hermitian quantum state over time given by *(Equation 6)* (see also [53]). Meanwhile, Ref. [36] (see also the earlier Ref. [54]) focuses on the applications of the entropy functional *(Equation 12)* to quantifying entanglement in non-Hermitian quantum systems, while also providing a relationship to negative central charges in non-unitary conformal field theories. Furthermore, Ref. [37] computes the Page curve associated with such an entropy. More recently, Ref. [55] analyzed the second moment of $\varrho_{AB}$ from *(Equation 15)*, which is a spatiotemporal generalization of the purity for density matrices.*

## 4. Dynamical Measures of Quantum Information

In this section, we employ the entropy of processes from (Equation 7) to define dynamical analogs of quantum conditional entropy, quantum mutual information, and a quantum analog of the conditional information loss for classical channels appearing in Ref. [13], which we refer to as *quantum information discrepancy*. After defining such dynamical measures of quantum information, we give various examples that highlight their meaning. We then close this section with a remark pointing out how the information measures defined in this section are related by an equation, which one may view as representing conservation of information for open quantum systems that interact with an environment.

**Definition** **2** (Dynamical measures of quantum information).
*Let (E,ρ) be a quantum process, so that E:A→B is a quantum channel and ρ∈S(A) is a state.*


*The **conditional entropy** of (E,ρ) is the real number H(E,ρ) given by*

H(E,ρ)=S(E,ρ)−S(ρ).


*The 
**mutual information** of (E,ρ) is the real number I(E,ρ) given by*

$$I\left(E,\rho \right)=S\left(\rho \right)+S\left(E(\rho )\right)-S\left(E,\rho \right)\;.$$


*The 
**information discrepancy**
 of (E,ρ) is the real number K(E,ρ) given by*

$$K\left(E,\rho \right)=S\left(E,\rho \right)-S\left(E(\rho )\right)\;.$$




**Remark** **3.**
*If E⋆ρ∈A⊗B is positive, then we may set $E⋆\rho =\rho_{AB}$, where $\rho_{AB}$ is a bipartite density operator representing the joint state of two spacelike-separated regions A and B whose associated marginals are $\rho_{A}=\rho$ and $\rho_{B}=E\left(\rho \right)$. In such a case, H(E,ρ)=H(B|A) is the usual quantum conditional entropy and I(E,ρ)=I(A:B) is the usual quantum mutual information of the joint state $\rho_{AB}$. States $\rho_{AB}$ that may also be viewed as quantum states over time will be referred to as **dual states.***


**Remark** **4.**
*If ρ is a pure state, i.e., if S(ρ)=0, then it follows that for any process (E,ρ) we have*

(16)
S(E,ρ)=H(E,ρ)andI(E,ρ)=−K(E,ρ).

*Thus, the information measures H(E,ρ) and K(E,ρ) are redundant in this case.*


We now give several examples that help clarify the physical meaning of such dynamical measures of quantum information.

**Example** **1** (The discard-and-prepare channel).
*Let E:A→B be the discard-and-prepare channel, which is given by E(a)=tr(a)σ for some state σ∈S(B). It then follows that E⋆ρ=ρ⊗σ for all ρ∈S(A). Hence,*

I(E,ρ)=S(ρ)+S(σ)−S(ρ⊗σ)=0,

*where the final equality follows from the additivity of von Neumann entropy. As such, the mutual information I(E,ρ)=0 for all states ρ, which is consistent with the interpretation of mutual information as a measure of correlation/shared information between systems, as there are no correlations established between the input and the output of E for all states ρ. Moreover, we have*

K(E,ρ)=S(ρ⊗σ)−S(σ)=S(ρ),

*which is consistent with the interpretation that K(E,ρ) quantifies the information loss/gain associated with the process (E,ρ). The fact that K(E,ρ) is positive for states ρ with positive von Neuman entropy indicates that information is in fact lost in the process (E,ρ), and the information that is lost is precisely S(ρ), as ρ has been discarded in the process (E,ρ). As for the conditional entropy, we have H(E,ρ)=S(σ), which is consistent with the interpretation that conditional entropy measures the uncertainty of the output given knowledge of the input. In this example, knowledge of the input state ρ does not affect in any way the uncertainty contained in the output state σ, which is then reflected in the fact that H(E,ρ)=σ.*


**Example** **2** (Unitary evolution).
*Let U:A→B be a unitary channel so that there exists some unitary U∈A such that $U\left(a\right)=UaU^{†}$ for all a∈A. By Lemma A5 in Appendix A, it follows that S(U,ρ)=S(ρ) for every state ρ∈S(A). Thus, H(U,ρ)=K(U,ρ)=0 and I(U,ρ)=S(ρ). The fact that H(U,ρ)=0 is consistent with the fact that unitary evolution is a deterministic process, and hence there is no uncertainty in the output of the process given knowledge of the input. Moreover, no information is lost or gained in a unitary process, which is reflected in the fact that K(U,ρ)=0. Finally, the shared information between ρ and U(ρ) is precisely the information contained in the input state ρ, resulting in a mutual information I(U,ρ)=S(ρ).*


**Example** **3** (Holevo information as dynamical mutual information).
*Let ${\left\{\rho_{i}\right\}}_{i=1}^{k}\subset S\left(B\right)$ be a collection of states, and let E:A→B be a classical-quantum channel given by $E(|i\rangle \langle j|)=\delta_{ij}\;\rho_{i}$, so that A is necessarily a k-dimensional system. Given a classical state $p=\sum_{i}p_{i}\left|i\rangle \langle i\right|$, the process (E,p) models a protocol in which Alice sends the state $\rho_{i}$ to Bob with probability $p_{i}$. In such a case, we have*

$$E⋆p=\sum_{i=1}^{k}p_{i}\left|i\rangle \langle i\right|⊗\rho_{i}\;,$$

*which is a dual state as E⋆p is a separable bipartite density matrix, and hence positive. We then have*

$$\begin{array}{cc} I(E,p) & =H\left(p\right)+S\left(\sum_{i}p_{i}\rho_{i}\right)-S\left(E⋆p\right)=H\left(p\right)+S\left(\sum_{i}p_{i}\rho_{i}\right)-S\left(\sum_{i=1}^{k}p_{i}\left|i\rangle \langle i\right|⊗\rho_{i}\right)\\ \; & =H\left(p\right)+S\left(\sum_{i}p_{i}\rho_{i}\right)-\left(H\left(p\right)+\sum_{i}p_{i}S(|i\rangle \langle i|⊗\rho_{i})\right)=S\left(\sum_{i}p_{i}\rho_{i}\right)-\sum_{i}p_{i}S\left(\rho_{i}\right)\;, \end{array}$$

*where the third equality follows from the strong convex linearity property of the entropy functional S (item (iv) of Proposition 1), and the final equality follows from the additivity of the entropy functional S (item (iii) of Proposition 1). The quantity χ given by*

$$\chi =S\left(\sum_{i}p_{i}\rho_{i}\right)-\sum_{i}p_{i}S\left(\rho_{i}\right)$$

*is referred to as the Holevo information associated with the ensemble $\left\{\left(p_{i},\rho_{i}\right)\right\}$, which is an upper bound for the accessible information Bob can extract from the ensemble ρ via positive operator-valued measurements (cf. [56] (Section 11.6.1)). As such, the fact that I(E,p)=χ yields a dynamical interpretation for the Holevo information χ, while also providing further justification for our use of the entropy functional S.*


**Example** **4** (Partial trace).
*Let E:A⊗B→B be the partial trace, suppose A and B are systems consisting of a single qubit, and let $\rho_{\mathrm{EPR}}\in A⊗B$ be the density matrix given by*

$$\rho_{\mathrm{EPR}}=\frac{1}{2}\left(\begin{array}{cccc} 0 & 0 & 0 & 0\\ 0 & 1 & -1 & 0\\ 0 & -1 & 1 & 0\\ 0 & 0 & 0 & 0 \end{array}\right)\;,$$

*which is a state corresponding to an EPR pair of maximally entangled qubits. As the multi-spectrum of $E⋆\rho_{\mathrm{EPR}}$ is given by*

$$\mathrm{mspec}\left(E⋆\rho_{\mathrm{EPR}}\right)=\left\{-1/4,-1/4,0,0,0,0,3/4,3/4\right\}\;,$$

*it follows that*

$$H\left(E,\rho_{\mathrm{EPR}}\right)=S\left(E,\rho_{\mathrm{EPR}}\right)=\log\left(\frac{4}{3\sqrt{3}}\right)\approx -0.3774<0\;,$$

*thus providing an example of negative conditional entropy. While the precise meaning of the negative conditional entropy in this example is less clear, its negativity is consistent with the fact that negative conditional entropy is a signature of entanglement, indicating that knowledge of the state $\rho_{B}=E\left(\rho_{\mathrm{EPR}}\right)$ enables one to infer knowledge about the initial state of the joint system as a whole [8,57,58,59,60,61].*


**Remark** **5** (Conservation of Quantum Information).*Note that for every process (E,ρ), we have*(17)S(ρ)=I(E,ρ)+K(E,ρ),*which we view as an equation representing the conservation of information for quantum processes (E,ρ). In particular, we view the LHS of Equation *(Equation 17)* as the information contained in system A in state ρ prior to an interaction with some environment system, which is modeled by the action of the quantum channel E:A→B on system A. The RHS of Equation *(Equation 17)* then indicates that the information contained prior to the interaction is dispersed post-interaction in the form of mutual information combined with information discrepancy. However, since K(E,ρ) can be negative, such a notion of conservation of information is distinctly quantum, as it may incorporate negative measures of information.*

## 5. On the Non-Negativity of Dynamical Mutual Information

In Remark 1, we showed the entropy functional S(X)=−tr(Xlog|X|) is not subadditive in general, even when restricted to quasi-states (i.e., unit trace Hermitian matrices) whose marginals are density matrices. However, the quasi-state we constructed that violated subadditivity was shown not to be a quantum state over time, leading us to conjecture that the entropy functional S(X)=−tr(Xlog|X|) does in fact satisfy subadditivity when restricted to quantum states over time. From the perspective of dynamical measures of quantum information, such a conjecture is equivalent to the conjecture that the dynamical mutual informationI(E,ρ)=S(ρ)+S(E(ρ))−S(E,ρ)
is non-negative for all processes (E,ρ).

In this section, we present numerical evidence in support of the conjecture that I(E,ρ)≥0 for all processes (E,ρ). This is achieved by sampling random states ρ and channels E according to the Haar measure for various dimensions [62]. The Haar-random density matrices ρ are obtained by first generating a Haar-random pure state |ψ〉 in $C^{m}⊗C^{d_{1}}$, and then ρ is defined via the partial trace as $\rho ={\mathrm{tr}}_{M_{d_{1}}}\left(|\psi \rangle \langle \psi |\right)$. The random channel E is then constructed in the following way. First, a Haar-random environment system state $\rho_{E}$ is constructed on $M_{d_{3}}$ by generating a Haar-random |φ〉 in $C^{d_{3}}⊗C^{d_{2}}$ and taking the partial trace $\rho_{E}={\mathrm{tr}}_{M_{d_{2}}}\left(|\varphi \rangle \langle \varphi |\right)$ ($M_{k}$ denotes the algebra of complex k×k matrices for all k>0). Then, a Haar-random unitary $U:C^{m}⊗C^{d_{3}}\to C^{m}⊗C^{d_{3}}$ is sampled. Finally, $E={\mathrm{tr}}_{M_{d_{3}}}\circ {\mathrm{Ad}}_{U}\circ j_{E}$, where $j_{E}:M_{m}\to M_{m}⊗M_{d_{3}}$ is the environment inclusion map sending an arbitrary state σ to $\sigma ⊗\rho_{E}$. Although this is not the most general quantum channel that one can construct, it nevertheless has a direct physical interpretation in terms of coupling a system, A, to a random environment and undergoing a random unitary evolution, followed by discarding (i.e., tracing out) the environment [62].

Figure 3 suggests that I(E,ρ)≥0 may indeed hold for general processes (E,ρ). If true, this would be in stark contrast to alternative proposals for dynamical forms of entropies, for which the subadditivity of the entropy functions is known to fail for quantum states over time [3,18]. Finally, we point out that if we assume the validity of the lower bound 0≤I(E,ρ), then the analog of the upper bound $I\left(A:B\right)\leq 2\min\left\{S\left(\rho_{A}\right),S\left(\rho_{B}\right)\right\}$ [1,57,63] must fail in our dynamical setting. This can easily be seen by taking any pure state ρ as the initial state, which forces $\min\left\{S(\rho ),S(E(\rho ))\right\}=0$, while I(E,ρ) is generically positive. Nevertheless, our numerical plots suggest the possibility that the alternative weaker bound $I\left(E,\rho \right)\leq 2\min\left\{\log\left(\mathrm{tr}\left(1_{A}\right)\right),\log\left(\mathrm{tr}\left(1_{B}\right)\right)\right\}$ may hold, in analogy with spatial quantum mutual information (cf. [56] (Exercise 11.6.3)).

## 6. The Information Gain Due to Measurement

Let ${\left\{M_{k}\right\}}_{k=1}^{n}$ be a positive operator-valued measure (POVM) on system *A* so that ${\left\{M_{k}\right\}}_{k=1}^{n}\subset A$ is a collection of positive operators such that $\sum_{k}M_{k}=1$. Such a POVM determines a quantum-classical channel E:A→B given by(18)$$E\left(\rho \right)=\sum_{k=1}^{n}\mathrm{tr}\left(\rho M_{k}\right)|k\rangle \langle k|\;,$$
which necessarily implies that *B* is an *n*-dimensional system. As the probability of the measurement outcome $M_{k}$ is $\mathrm{tr}(\rho M_{k})$, the state q=E(ρ)∈S(B) is a classical state encoding the probabilities of the measurement outcomes associated with the POVM ${\left\{M_{k}\right\}}_{k=1}^{n}$. In such a case, we refer to the process (E,ρ) as a ***measurement process*** for all ρ∈S(A).

In this section, we show that if ρ∈S(A) is a pure state, then the information discrepancy K(E,ρ) associated with the measurement process (E,ρ) is necessarily non-positive and K(E,ρ) may be interpreted as quantifying the disturbance of the state ρ due to the measurement of the POVM ${\left\{M_{k}\right\}}_{k=1}^{n}$. For this, we first derive a formula for K(E,ρ) in the case that (E,ρ) is a measurement process.

**Proposition** **2.***Let ${\left\{M_{k}\right\}}_{k=1}^{n}$ be a POVM on system A, let E:A→B be the quantum-classical channel given by *(Equation 18)*, let ρ∈S(A) be a state, and let $\rho_{k}\in A$ be the element given by*(19)$$\rho_{k}=\frac{\rho M_{k}+M_{k}\;\rho }{2q_{k}}\;,$$*where $q_{k}=\mathrm{tr}\left(\rho M_{k}\right)$ for all k∈{1,…,n}. Then, $K\left(E,\rho \right)=\sum_{k=1}^{n}q_{k}S\left(\rho_{k}\right)$.*

**Proof.** Let $E_{ij}^{A}=\left|i\rangle \langle j\right|$ denote the matrix units in A, let $E_{kl}^{B}=\left|k\rangle \langle l\right|$ denote the matrix units in B, and let $M_{k}^{ij}=\langle i|M_{k}|j\rangle$ for all *i* and *j*. We then have$$J\left[E\right]=\sum_{i,j}E_{ij}^{A}⊗E\left(E_{ji}^{A}\right)=\sum_{i,j}E_{ij}^{A}⊗\left(\sum_{k}\mathrm{tr}\left(E_{ji}^{A}M_{k}\right)E_{kk}^{B}\right)=\sum_{i,j,k}M_{k}^{ij}E_{ij}^{A}⊗E_{kk}^{B}=\sum_{k}M_{k}⊗E_{kk}^{B}\;.$$Thus,$$E⋆\rho =\frac{1}{2}\left\{\rho ⊗1\;,J[E]\right\}=\sum_{k}q_{k}\left(\rho_{k}⊗E_{kk}^{B}\right)\;.$$The strong convex linearity of entropy (item iv. of Proposition 1) then yields$$S\left(E⋆\rho \right)=H\left(q\right)+\sum_{k}q_{k}S\left(\rho_{k}\right)\;,$$
where H(q) denotes the Shannon entropy of the distribution $q_{k}$. Moreover, since$$S\left(E(\rho )\right)=S\left(\sum_{k}q_{k}E_{kk}^{B}\right)=H\left(q\right)+\sum_{k}q_{k}S\left(E_{kk}^{B}\right)=H\left(q\right)\;,$$
we have$$K\left(E,\rho \right)=S\left(E⋆\rho \right)-S\left(E\left(\rho \right)\right)=H\left(q\right)+\sum_{k}q_{k}S\left(\rho_{k}\right)-H\left(q\right)=\sum_{k}q_{k}S\left(\rho_{k}\right)\;,$$
as desired. □

We now give a precise definition of a disturbing/non-disturbing measurement, which we will show is naturally quantified by K(E,ρ).

**Definition** **3.**
*A measurement process (E,ρ) with ρ pure is said to be **non-disturbing** if and only if im(ρ) is invariant with respect to the POVM elements ${\left\{M_{k}\right\}}_{k=1}^{n}$ associated with E for all k∈{1,…,n}. Otherwise, the measurement process (E,ρ) is said to be **disturbing.***


**Remark** **6.**
*The definition of a non-disturbing process in Definition 3 is in agreement with the notion of a non-disturbing measurement on a state given in Section 9.4 of Ref. [56] provided that we employ the square-root instrument associated with a POVM E with POVM elements ${\left\{M_{k}\right\}}_{k=1}^{n}$. This can be seen by the fact that if ρ=|v〉〈v| is a pure state, then the assumption that im(ρ) is invariant with respect to $M_{k}$ means that $M_{k}|v\rangle =c_{k}|v\rangle$ for some $c_{k}>0$, which implies $\sqrt{M_{k}}|v\rangle =\sqrt{c_{k}}|v\rangle$. Hence, $\sqrt{M_{k}}\;\rho \sqrt{M_{k}}=c_{k}\rho$ so that the updated state via the state-update rule for the square-root instrument (cf. Section IV.B. of Ref. [33]) equals ρ (this corresponds to the case ϵ=0 in Equation (9.197) of Ref. [56]), in which case ρ is undisturbed.*


**Theorem** **2.**
*Let (E,ρ) be a measurement process with pure ρ. Then, the following statements hold.*

*i.* 
*If (E,ρ) is non-disturbing, then K(E,ρ)=0.*
*ii.* 
*If (E,ρ) is disturbing, then K(E,ρ)<0.*



**Proof.** By choosing a basis, we may assume that the Hilbert space of the system under consideration is $C^{m}$ for some m>0. So let $|v\rangle \in C^{m}$ be such that ρ=|v〉〈v|, let $\left\{M_{k}\right\}$ be the POVM elements associated with E, and let $\rho_{k}$ be the elements as given by (Equation 19) for all *k*, so that for all $|w\rangle \in C^{m}$, we have(20)$$\rho_{k}|w\rangle =\frac{\langle v|M_{k}|w\rangle }{2q_{k}}|v\rangle +\frac{\langle v|w\rangle }{2q_{k}}M_{k}|v\rangle \;,$$
where we recall that $q_{k}=\mathrm{tr}\left(\rho M_{k}\right)$ for all *k*. It then follows that $\mathrm{im}\left(\rho_{k}\right)\subseteq \mathrm{span}\left\{|v\rangle ,M_{k}|v\rangle \right\}$, so $\rho_{k}$ is a self-adjoint matrix with a rank of at most 2 for all *k*.
Suppose (E,ρ) is non-disturbing so that im(ρ) is invariant with respect to $M_{k}$ for all *k*. It then follows from (Equation 20) that $\rho_{k}$ is a rank-1 projection since $\mathrm{tr}(\rho_{k})=1$. Therefore, $S(\rho_{k})=0$ for all *k*. Thus, K(E,ρ)=0 by Proposition 2.Suppose (E,ρ) is disturbing so that there exists a *k* such that im(ρ) is *not* invariant with respect to $M_{k}$. It then follows from (Equation 20) that $\rho_{k}$ is of rank 2. We now show that $\rho_{k}$ is not positive by determining its non-zero eigenvalues. For this purpose, let $|u\rangle =M_{k}|v\rangle$, so that $\rho_{k}=\frac{1}{2q_{k}}\left(|v\rangle \langle u|+|u\rangle \langle v|\right)$ and $\langle v|u\rangle =\langle u|v\rangle =\langle v|M_{k}|v\rangle =q_{k}$. Writing an arbitrary eigenvector of $\rho_{k}$ as |w〉=α|v〉+β|u〉, with α,β∈C, the eigenvalue equation $\rho_{k}|w\rangle =\lambda |w\rangle$ yields$$\lambda \alpha |v\rangle +\lambda \beta |u\rangle =\frac{\alpha q_{k}+\beta \langle u|u\rangle }{2q_{k}}|v\rangle +\frac{\alpha +\beta q_{k}}{2q_{k}}|u\rangle \;.$$By the linear independence of {|v〉,|u〉}, this guarantees$$\lambda \alpha =\frac{\alpha q_{k}+\beta \langle u|u\rangle }{2q_{k}}\;\;\mathrm{and}\;\;\lambda \beta =\frac{\alpha +\beta q_{k}}{2q_{k}}.$$These equations then yield the quadratic equation$$4\lambda^{2}-4\lambda +1-\frac{\langle u|u\rangle }{q_{k}^{2}}=0\;,$$
whose solutions $\lambda_{\pm }$ are given by$$\lambda_{\pm }=\frac{1}{2}\pm \frac{\langle u|u\rangle }{2q_{k}}.$$Note that $\lambda_{+}+\lambda_{-}=1$, as expected. Now let $\{|v\rangle ,|v_{1}\rangle ,\cdots ,|v_{n-1}\rangle \}$ be an orthonormal basis of $C^{m}$ containing |v〉. By using the completeness relation$$1_{m}=\left|v\rangle \langle v\right|+\sum_{i=1}^{n-1}|v_{i}\rangle \langle v_{i}|\;,$$
we find that$$\begin{array}{cc} \langle u|u\rangle & =\langle v|M_{k}M_{k}|v\rangle =\langle v|M_{k}\left|v\rangle \langle v\right|M_{k}|v\rangle +\sum_{i=1}^{n-1}\langle v|M_{k}|v_{i}\rangle \langle v_{i}|M_{k}|v\rangle \\ \; & =q_{k}^{2}+\sum_{i=1}^{n-1}\left|\langle v\right|M_{k}|v_{i}\rangle {|}^{2}>q_{k}^{2}\;, \end{array}$$
which implies $\lambda_{-}<0$ and $\lambda_{+}>1$. Therefore,$$S\left(\rho_{k}\right)=-\lambda_{-}\log\left|\lambda_{-}\right|-\lambda_{+}\log\left(\lambda_{+}\right)<0$$(cf. Figure 1). This argument, together with the proof of item i, yields $S(\rho_{k})=0$ whenever im(ρ) is invariant with respect to $M_{k}$ and $S(\rho_{k})<0$ whenever im(ρ) is *not* invariant with respect to $M_{k}$. It then follows from Proposition 2 that if (E,ρ) is disturbing, then $K_{1}\left(E,\rho \right)<0$, as desired. □


In light of Theorem 2, we find that K(E,ρ) serves as a natural measure of disturbance associated with a measurement process (E,ρ). Moreover, as disturbing measurements are precisely the measurements that yield information, the fact that K(E,ρ) is negative for disturbing measurement processes suggests the interpretation that information is in fact *created* by the process of measurement, as we recall that a positive *K* indicates a loss of information (cf. Example 1). We emphasize that such information creation is a purely quantum phenomenon, since at the classical level, the information discrepancy *K* is always non-negative and hence is a measure of information loss [13].

## 7. Concluding Remarks

In this work, we defined the dynamical entropy S(E,ρ) associated with quantum processes (E,ρ), where ρ is a state and E is a CPTP map responsible for the dynamical evolution of ρ. We then used this entropy S(E,ρ) to define dynamical analogs of the quantum conditional entropy, the quantum mutual information, and an information measure we refer to as “information discrepancy”. Key to our formulation of dynamical entropy was an extension of von Neumann entropy to arbitrary Hermitian matrices using the real part of the analytic continuation of the logarithm to the negative real axis. This entropy function also yields a well-defined notion of entropy for quasi-probability distributions, which play a prevalent role in quantum theory [64,65,66,67,68].

We have shown that our information measures satisfy many properties of classical information measures associated with stochastic processes, such as the vanishing of conditional entropy under deterministic evolution. While classical information measures are always non-negative, the information measures defined in this work may be negative, which is a characteristic feature of quantum information versus classical information (see Table 1 for details). Our dynamical mutual information, however, seems to be an exception, since we find it to be non-negative in all known examples. This was in fact unexpected as our generalized entropy function does *not* satisfy subadditivity on quasi-density matrices, contrary to the case of the usual von Neumann entropy functional for density matrices. A proof that our dynamical mutual information is in fact a non-negative measure of quantum information still eludes us and thus remains an open problem. If our dynamical mutual information is in fact non-negative and bounded from above (as suggested by Figure 3), then one may define a form of channel capacity in analogy with the classical case by maximizing the mutual information of a channel over the set of input states. It would then be interesting to compare such a notion of channel capacity with more standard notions of channel capacity from quantum information theory [56,63,69,70,71,72].

While we have extensively studied such dynamical measures of quantum information from a mathematical viewpoint, their general operational interpretations are still lacking at this point (however, see Remark 5 and Theorem 2 for partial progress in this direction). It would also be desirable to find a more explicit and palatable connection between such information measures and fundamental aspects of quantum dynamics, such as causal correlations and entanglement in time [22]. This is particularly relevant now, as a theory of quantum states over time has recently been developed in order to study these types of fundamental questions [17,33].

## Figures and Tables

**Figure 1 entropy-27-00331-f001:**
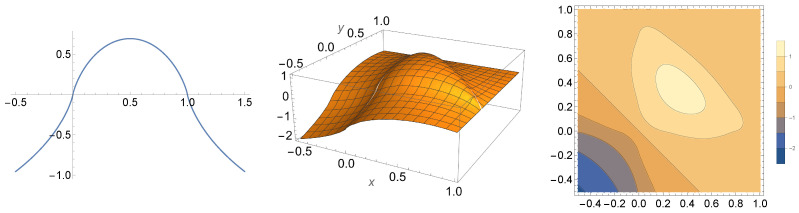
The left plot shows the graph of the function −xlog|x|−(1−x)log|1−x| for x∈[−1/2,3/2], which is what the entropy from Definition 1 looks like for *X*, a diagonal matrix of the form X=diag(x,1−x) whose diagonal entries form a quasi-probability distribution with entries contained in the interval [−1/2,3/2]. The middle plot shows the graph of the function −xlog|x|−ylog|y|−(1−x−y)log|1−x−y| for (x,y)∈[−1/2,1]×[−1/2,1]. The right plot shows the same information as the middle plot drawn as a contour plot.

**Figure 2 entropy-27-00331-f002:**
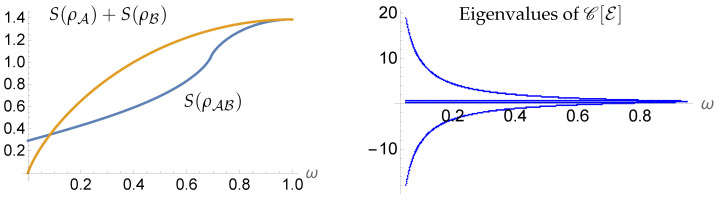
For the parameterized family of quasi-states in Remark 1, there exists an interval of values ω>0 such that subadditivity of the extended entropy fails (**left**). However, for the values of ω for which the extended entropy fails to be subadditive, the quasi-state $\rho_{AB}$ is not a quantum state over time. Indeed, the eigenvalues of the Choi matrix C[E] of the map E:A→B that would induce the dynamics via $\rho_{AB}=E⋆\rho_{A}$ are negative for a range of values of ω (**right**).

**Figure 3 entropy-27-00331-f003:**
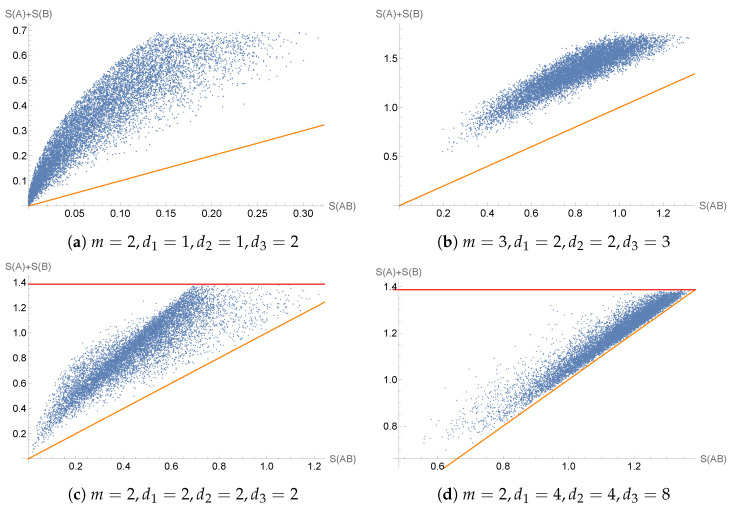
In these scatter plots, 10,000 Haar-random random input density matrices ρ∈A and quantum channels E:A→B are randomly sampled, with $A=B=M_{m}$, for some positive integer value of *m*. The data points on these plots are given by $\left(S(AB),S(A)+S(B)\right)=\left(S(E,\rho ),S(\rho )+S(E(\rho ))\right)$. The orange diagonal line depicts the line for which these two values would be equal. In other words, points above the line depict states ρ and channels E for which sub-additivity S(E,ρ)≤S(ρ)+S(E(ρ)) holds. Meanwhile, the red horizontal line is given by 2log(m) (this horizontal line is not shown in the scatter plots (**a**,**b**) because it lies well above the data points shown). In all plots, not a single violation of $0\leq I\left(E,\rho \right)\leq 2\min\left\{\log\left(\mathrm{tr}\left(1_{A}\right)\right),\log\left(\mathrm{tr}\left(1_{B}\right)\right)\right\}$ appears. The dimensions $m,d_{1},d_{2},d_{3}$ are different in each plot and are provided under the respective figures.

**Table 1 entropy-27-00331-t001:** (On the non-negativity of information measures) This table depicts the four measures of information studied in this paper: entropy, conditional entropy, mutual information, and information discrepancy (called “conditional information loss” in the classical setting [13]). In the static setting, the given datum is a joint state. In the dynamic setting, the given datum is a process (an initial state together with a channel). In the present paper, we have introduced the information measures in the column “quantum dynamic.” A check mark (✓) indicates that positivity holds for the information measure, while a cross mark (×) indicates that positivity does not hold in general for that information measure. Note that the positivity of the dynamical mutual information is conjectural at present, which is why “✓” is written in that entry.

	Classical Static	Classical Dynamic	Quantum Static	Quantum Dynamic
entropy	✓	✓	✓	×
conditional entropy	✓	✓	×	×
mutual information	✓	✓	✓	✓?
information discrepancy	✓	✓	×	×

## Data Availability

No new data were created or analyzed in this study.

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
