# Peer review of "On Dynamical Measures of Quantum Information"

_entropy, 2025, doi:10.3390/e27040331_

Round 1

Reviewer 1 Report

Comments and Suggestions for Authors

The manuscript entitled "On Dynamical Measures of Quantum Information" presents a spacetime generalization of von Neumann entropy. The proposed "entropy of a quantum process" is based on the framework of quantum states over time, which contains information about both the quantum state and the quantum channel and can recover the von Neumann entropy of states with trivial evolution. The authors then proceed to investigate the properties of the proposed measure and introduce dynamical analogs of quantum conditional entropy, mutual information, and the so-called information discrepancy. The authors also discuss its relation to measurement.

The work is mathematically rigorous, well-organized, and addresses a gap in the quantum states-over-time community. More generally, it contributes to the study of extending entropy to non-positive operators and should therefore be of interest to the readers of Entropy. In summary, I recommend the manuscript for publication in Entropy.

Below are some references I think should be mentioned in this work.

1) arXiv:2303.10544 gives a way to construct the quantum state over time.
2) PRL 134 (4), 04020(2025)1 has an analysis of the second moments of quantum state over time, should be useful to estimate the proposed entropy.
3) Nature 474, 61–63 (2011) gives a physical interpretation of negative conditional entropy which is relevant with example 28.

I have three comments on the presentation.   1) The authors have confused the labeling of equations with that of theorems and lemmas. 2) To prevent the reader from having to refer to the appendix for the lemma, the authors could include some lemmas (e.g., Lemma 40) in the main body while keeping their proofs in the appendix. 3) For consistency, the notation in the appendix should also follow Dirac notation.

Author Response

Reviewer 1:

Below are some references I think should be mentioned in this work.

1) arXiv:2303.10544 gives a way to construct the quantum state over time.
2) PRL 134 (4), 04020(2025)1 has an analysis of the second moments of quantum state over time, should be useful to estimate the proposed entropy.
3) Nature 474, 61–63 (2011) gives a physical interpretation of negative conditional entropy which is relevant with example 28.

 I have three comments on the presentation.   1) The authors have confused the labeling of equations with that of theorems and lemmas. 2) To prevent the reader from having to refer to the appendix for the lemma, the authors could include some lemmas (e.g., Lemma 40) in the main body while keeping their proofs in the appendix. 3) For consistency, the notation in the appendix should also follow Dirac notation.

Response: We have added the references suggested by the referee. As for the labeling, we have used this convention in prior articles published in Entropy. If Entropy editorial staff wishes to change it, they can. The appendix only essentially contains one result, namely, the determination of the eigenvalues of a state over time associated with a unitary process. We then broke the proof up into several lemmas to make the proof more readable. As such, it wouldn't really make sense to put a single Lemma from the appendix into the main text, as the lemmas are all used in sequence for the result about the eigenvalues of a state over time associated with a unitary process. For these reasons, we feel it is better to leave all the lemmas in the appendix. We have also included Dirac notation in the Appendix as well (see the proof of Lemma A1). 

Reviewer 2 Report

Comments and Suggestions for Authors

See the pdf file.

Author Response

We have incorporated the reviewer's comment regarding the proof of Theorem 36. We thank the reviewer for their detailed reading of our paper and correcting this error on our part.

Reviewer 3 Report

Comments and Suggestions for Authors

The present manuscript defines and discusses the properties of an extension of the von Neumann entropy to quantum states over time. It is shown that this new definition reduces to the standard von Neumann entropy for regular quantum states, and that it possess many appealing properties (generalized from from the standard von Neumann entropy). This includes extensions of entropy-like measures, such as conditional and mutual information.

The narrative is clear and convincing. The discussion is comprehensive and sound. Therefore, I can recommend the manuscript to be accepted for publication essentially in its current form.

There is only one minor issue: On page 8, something went wrong with the formatting of the itemized lists.

Author Response

There is only one minor issue: On page 8, something went wrong with the formatting of the itemized lists.

Response: We have fixed this formatting issue.